# Comparison of Primary Virus Isolation in Pulmonary Alveolar Macrophages and Four Different Continuous Cell Lines for Type 1 and Type 2 Porcine Reproductive and Respiratory Syndrome Virus

**DOI:** 10.3390/vaccines9060594

**Published:** 2021-06-03

**Authors:** Jiexiong Xie, Nick Vereecke, Sebastiaan Theuns, Dayoung Oh, Nathalie Vanderheijden, Ivan Trus, Jannes Sauer, Philip Vyt, Caroline Bonckaert, Christian Lalonde, Chantale Provost, Carl A. Gagnon, Hans Nauwynck

**Affiliations:** 1Laboratory of Virology, Faculty of Veterinary Medicine, Ghent University, 9820 Merelbeke, Belgium; xjx297@gmail.com (J.X.); nick.vereecke@ugent.be (N.V.); Dayoung.oh@ugent.be (D.O.); Nathalie.Vanderheijden@ugent.be (N.V.); 2PathoSense BV, 2500 Lier, Belgium; sebastiaan.theuns@pathosense.com (S.T.); jannes.sauer@pathosense.com (J.S.); 3Vaccine and Infectious Disease Organization (VIDO)—International Vaccine Centre, University of Saskatchewan, Saskatoon, SK S7N 5E3, Canada; ivan.trus@gmail.com; 4Dialab, Belsele, 9111 Sint-Niklaas, Belgium; philip.vyt@gmail.com; 5Animal Health Care Flanders (DGZ), 8820 Torhout, Belgium; caroline.bonckaert@dgz.be; 6Swine and Poultry Infectious Diseases Research Center (CRIPA)—Fonds de Recherche du Québec, Faculté de Médecine Vétérinaire, Université de Montréal, St-Hyacinthe, QC J2S 2M2, Canada; christian.lalonde@umontreal.ca (C.L.); chantale.provost@umontreal.ca (C.P.); carl.a.gagnon@umontreal.ca (C.A.G.)

**Keywords:** PRRSV, virus isolation/production, macrophages, MARC-145, PK15^Sn-CD163^, PK15^S10-CD163^

## Abstract

Porcine Reproductive and Respiratory Syndrome Virus (PRRSV) has a highly restricted cellular tropism. In vivo, the virus primarily infects tissue-specific macrophages in the nose, lungs, tonsils, and pharyngeal lymphoid tissues. In vitro however, the MARC-145 cell line is one of the few PRRSV susceptible cell lines that are routinely used for in vitro propagation. Previously, several PRRSV non-permissive cell lines were shown to become susceptible to PRRSV infection upon expression of recombinant entry receptors (e.g., PK15^Sn-CD163^, PK15^S10-CD163^). In the present study, we examined the suitability of different cell lines as a possible replacement of primary pulmonary alveolar macrophages (PAM) cells for isolation and growth of PRRSV. The susceptibility of four different cell lines (PK15^Sn-CD163^, PK15^S10-CD163^, MARC-145, and MARC-145^Sn^) for the primary isolation of PRRSV from PCR positive sera (both PRRSV1 and PRRSV2) was compared with that of PAM. To find possible correlations between the cell tropism and the viral genotype, 54 field samples were sequenced, and amino acid residues potentially associated with the cell tropism were identified. Regarding the virus titers obtained with the five different cell types, PAM gave the highest mean virus titers followed by PK15^Sn-CD163^, PK15^S10-CD163^, MARC-145^Sn^, and MARC-145. The titers in PK15^Sn-CD163^ and PK15^S10-CD163^ cells were significantly correlated with virus titers in PAM for both PRRSV1 (*p* < 0.001) and PRRSV2 (*p* < 0.001) compared with MARC-145^Sn^ (PRRSV1: *p* = 0.22 and PRRSV2: *p* = 0.03) and MARC-145 (PRRSV1: *p* = 0.04 and PRRSV2: *p* = 0.12). Further, a possible correlation between cell tropism and viral genotype was assessed using PRRSV whole genome sequences in a Genome-Wide-Association Study (GWAS). The structural protein residues GP2:187L and N:28R within PRRSV2 sequences were associated with their growth in MARC-145. The GP5:78I residue for PRRSV2 and the Nsp11:155F residue for PRRSV1 was linked to a higher replication on PAM. In conclusion, PK15^Sn-CD163^ and PK15^S10-CD163^ cells are phenotypically closely related to the in vivo target macrophages and are more suitable for virus isolation and titration than MARC-145/MARC-145^Sn^ cells. The residues of PRRSV proteins that are potentially related with cell tropism will be further investigated in the future.

## 1. Introduction

Porcine reproductive and respiratory syndrome (PRRS) is one of the most economically devastating swine diseases worldwide. PRRS virus (PRRSV), the causative agent of PRRS, is a small enveloped positive-stranded RNA virus that belongs to the *Arteriviridae* family in the order of the *Nidovirales*. The virus shows a high level of genetic diversity. Based on current genetic analysis, the traditional Genotype 1 (European genotype) and Genotype 2 (American genotype) have been reclassified into two distinct virus species namely *Betaarterivirus suid 1* and *Betaarterivirus suid 2* [1]. Three (potentially four) subgenotypes have been identified within *Betaarterivirus suid 1*. While all of these subtypes are present in Eastern Europe, subgenotype 1 is the predominant genotype circulating in Western and Central Europe. Therefore, subgenotypes 2, 3, and 4 are typically referred to as “East European subtypes” [2]. Recent Belgian PRRSV isolates still belong to subgenotype 1 but demonstrate increased pathogenicity as well as an altered macrophage tropism due to a changed use of entry mediators [3,4,5]. Type 2 PRRS viruses are classified into nine distinct lineages based on the global Open Reading Frame 5 (ORF5) sequence analysis [6]. Within these lineages, lineages 1 and 2 are mainly distributed in Canada and the United States (US) and lineages 3 and 4 are mostly distributed in Asia. Lineage 5, containing Ingelvac PRRS MLV vaccine, is the most cosmopolitan. Lineages 6–9 are also called US-like lineages, as most of the isolates are from the US. The co-existence of different (sub)genotypes, recombination events, new emerging isolates with increased virulence, and broadened cell tropism make PRRSV one of the most feared viruses in the global pig industry [3,5].

The genome of PRRSV consists of a single-stranded positive-sense RNA containing a 5′ cap and a 3′ poly (A) tail [7]. The genome (approximately 15.0 kb) contains at least 11 known open reading frames (ORFs). The replicase gene consists of ORF1a and ORF1b encoding two large nonstructural polyproteins, pp1a, and pp1ab. Both polyproteins are post-translationally processed into at least 14 non-structural proteins. The region downstream ORF1a and ORF1b contains several smaller, partially overlapping ORFs that encode the structural proteins of PRRSV. ORF2a, 2b, 3, 4, and 5a encode the structural proteins GP2, E, GP3, GP4, and ORF5a. ORF 5, 6, and 7 encode the structural proteins GP5, M, and N proteins. GP2/3/4 form a disulphide-linked heterotrimeric complex in virus particles. It was postulated that GP4 is critical for mediating inter-glycoprotein interactions and, along with GP2a, interacts with CD163, a step that is important for virus disassembly in macrophages [8]. GP2a has also been proven to be important for the adaptation of PRRSV to MARC-145 cells [9,10]. GP5 is the major glycoprotein of PRRSV. The covalent association of GP5 with M is crucial for virus assembly. The GP5-M complex interacts with heparin [11] and sialoadhesin [12], which mediates the internalization of the virus into the host cell, the macrophage. The N(ucleocapsid) protein of PRRSV is the most abundant and multifunctional protein. It is supposed to modulate nucleic acid binding and protein-protein interactions [13]. The N protein interacts with itself through covalent and non-covalent interactions [14]. The N-terminal domain of the protein contains many positively charged residues and presumably binds RNA [15]. A previous study demonstrated that the nucleocapsid protein can regulate viral RNA synthesis through interaction with nsp9 and cellular DHX9 [16].

PRRSV has a highly restricted cellular tropism. *In vivo*, the virus primarily infects tissue-specific macrophages in the nose, lungs, tonsils, and pharyngeal lymphoid tissues [17,18,19,20]. Long-term replication in superficial macrophages in mucosae may facilitate efficient virus transmission via oral and nasal secretions. In vitro, some PRRSV isolates were also found to replicate in a non-porcine cell line that does not belong to the monocyte/macrophage lineage, namely the African green monkey kidney cell line MA-104 and cells derived thereof (MARC-145 and CL2621) [21,22]. The presence of cellular receptors determines the susceptibility of cells for a PRRSV infection. At least seven molecules have been described as entry mediators for PRRSV: heparan sulphate (HS), Sialoadhesin (Sn, Siglec-1, CD169), Siglec-10, CD163, CD151, vimentin, DC-SIGN (CD209), and MYH9 [4,23,24,25,26,27]. Several cell lines, non-permissive for a PRRSV infection, such as BHK-21, PK15, and CHO-K1, have been shown to become susceptible to PRRSV infection upon introduction and expression of the recombinant receptor proteins. Previously, we have established two stably transfected cell lines: one expressing both Siglec-1 (= sialoadhesin) and CD163 (PK15^Sn-CD163^) and one cell line stably expressing both Siglec-10 and CD163 (PK15^S10-CD163^) [5,28]. Both of them were identified to be an optimal replacement for both MARC-145 and PAM for PRRSV virus isolation as well as virus production. The observed variability in growth characteristics of different virus strains tested in both cell lines is suggested to be related to the receptor usage [5,28]. Highly susceptible cell lines may be of use to make autogenous vaccines [29]. To determine the suitability of using different cells as a possible replacement for PAM cells in the isolation and growth of PRRSV, we compared the susceptibility of PAM with that of four different cell types: PK15^Sn-CD163^, PK15^S10-CD163^, MARC-145, and MARC-145^Sn^ (a MARC-145 cell line stably expressing sialoadhesin) for the primary isolation of PRRSV (both PRRSV1 and PRRSV2). In addition, it was examined if there is a correlation between the cell tropism and viral genotype.

## 2. Materials and Methods

### 2.1. Cells and Field Samples

Primary PAM were obtained from 4-to-6-week-old conventional pigs from a PRRSV-negative herd, as described previously [30]. Cells were maintained in Roswell Park Memorial Institute (RPMI) medium supplemented with 10% Fetal Bovine Serum (FBS), 2 mM L-glutamine, 1 mM sodium pyruvate, 1% of non-essential amino acids, and a mixture of antibiotics. MARC-145 cells, MARC-145^Sn^, PK15^Sn-CD163^, and PK15^S10-CD163^ cells were cultivated in Modified Eagle Medium (MEM), supplemented with 10% FBS, 100 U/mL penicillin, and 0.1 mg/mL streptomycin. Field serum samples were collected from PRRSV positive farms. PRRSV1 positive samples were collected in Belgian farms. PRRSV2 positive serum samples were collected in Canadian farms.

### 2.2. Virus Titration

In total, 77 serum samples defined to be PCR positive were collected in 2018–2019. Virus titration was performed on PAM, PK15^Sn-CD163^ and PK15^S10-CD163^, MARC-145 cells, and MARC-145^Sn^ cells as described previously [31]. Briefly, f or virus titration of the serum samples, PAM, PK15^Sn-CD163^ and PK15^S10-CD163^, MARC-145 cells, MARC-145^Sn^ were cultivated and subsequently inoculated with 10-fold dilutions of the samples. After 72 h, cells were fixed and virus-infected cells were stained with PRRSV-specific monoclonal antibodies (13E2, IgG_2a_) against the nucleocapsid protein [32]. This monoclonal antibody recognizes both PRRSV1 and PRRSV2. Virus titers were calculated by the method of Reed and Muench [33] and expressed as log_10_ median tissue culture infectious dose (TCID_50_) per mL.

### 2.3. Virus Sequencing and Phylogenetic Analysis

For PRRSV1 field strains, third generation nanopore sequencing was used following an in-house developed protocol by PathoSense BV (Merelbeke, Belgium). In short, viral RNA was extracted from isolated viruses using the Quick-DNA/RNA viral kit (Zymo Research), after which a PRRSV-specific enrichment was performed prior to library preparation (LSK-109; ONT) and sequencing on a new MinION flow cell (R9.4.1; ONT). The resulting raw data were basecalled, demultiplexed, and filtered using Guppy (v3.6; ONT), qcat (v1.1.0; ONT), and NanoFilt (v2.7.1) [34], respectively. De novo PRRSV1 genomes were generated using Canu (v2.0) [35], graphmap (v0.5.2) [36], and medaka polishing (v1.0.0; ONT). Sequencing of the wild-type PRRSV2 strains was done using the protocol reported earlier [37,38]. In brief, viral RNA was isolated from PRRSV qPCR positive clinical samples with a poly(A)-tail mRNA magnetic isolation module (New England BioLabs), followed by in vitro synthesis of dsDNA (New England BioLabs), sequencing libraries preparation, purification, and normalization (Illumina Nextera XT DNA library preparation kit). Sequencing libraries were sequenced in a v3 600-cycle cartridge using an Illumina MiSeq instrument, and PhiX was included at ~1% level to the total sequencing libraries as a control to establish the efficacy of the sequencing run. The genomes were assembled using de novo alignment and, when necessary, by alignment of the reads against strains from an in-house database, both using CLC genomics workbench (QIAGEN). Downstream analyses included whole genome alignments using the MAFFT web bioinformatics framework (https://mafft.cbrc.jp/alignment/software/, accessed on 1 November 2020) [39]. The recombination analysis (RIP, http://hiv.lanl.gov, accessed on 1 November 1995) was used to identify possible recombination points within the PRRSV genome, and a phylogenetic tree was constructed using IQ-Tree software (v1.5.5) (http://www.iqtree.org/ accessed on 15 August 2019). The constructed tree was presented and edited with FigTree V1.4.4 software (v1.4.4) (https://github.com/rambaut/figtree/releases accessed on 26 November 2018).

### 2.4. Correlation Analysis between Cell Tropism Phenotypes and Viral Structural Protein Sequences

We analyzed the existence of a possible relationship between the cell tropism phenotypes and viral genome sequences. The genome sequences of the PRRSV1 (*n* = 24) and PRRSV2 (*n* = 30) strains were sorted into two separate groups based on the titration results. The first group contained MARC-145 permissive strains and the second group, non-permissive strains. A Genome-Wide Association Study (GWAS) was performed based on these two groups. The difference between the non-MARC 145 and MARC-145 grower strains was assessed using the custom R script (https://github.com/itrus/GWAS-fasta accessed on 22 February 2015) [40]. The α = 0.01 level for the chi-square test was taken to denote the statistical significance in between-group comparisons. Similarly, strains with high titers (≥3log_10_TCID_50_) and low titers (<3log_10_TCID_50_) on PAM were compared and analyzed. The results were illustrated using a Manhattan dot plot. Significantly different (*p* < 0.01 or < 0.001) residues between the analyzed groups were labeled and identified within the viral genome. They were plotted together with the phylogenetic tree and titration results.

### 2.5. Statistical Analysis

Differences in titers for all tested samples using five cell types were analyzed by SPSS. Differences among cell types and virus genotypes were analyzed separately by one-way analysis of variance (ANOVA) followed by Turkey’s multiply comparison test. Differences were considered significant when *p* < 0.05. Correlation regression analysis of titers obtained with PAM and the titers obtained with other cell types were processed with Pearson’s correlation test. A correlation was considered significant when *p* < 0.05.

## 3. Results

### 3.1. Titration Results of the Serum Samples on PAM, PK15^Sn-CD163^, PK15^S10-CD163^, MARC-145^Sn^ and MARC-145

PCR positive serum samples from the pig farms were titrated simultaneously on PAM, PK15^Sn-CD163^, PK15^S10-CD163^, MARC-145, and MARC-145^Sn^ cell cultures. Titration results are summarized in Table 1. In total, 77 samples (47 of PRRSV1 and 30 of PRRSV2 samples) were tested. 62.3% of the samples were found to be positive with PAM, 46.8% on PK15^Sn-CD163^, 39.0% on PK15^S10-CD163^, 28.6% on MARC-145^Sn^ cells, and 27.3% on MARC-145 cells. Overall, growth of PRRSV1 field samples was more difficult on all tested cell types in contrast to PRRSV2. MARC-145 and MARC-145^Sn^ cell lines showed a significantly lower sensitivity towards PRRSV1 samples as compared with PRRSV2 isolates (10.6% vs. 56.7% and 14.9% vs. 46.7%, respectively). Both PK15^Sn-CD163^ and PK15^S10-CD163^ showed a relatively higher sensitivity for both types of PRRSV compared with MARC-145 and MARC-145^Sn^ cells but a lower sensitivity compared with PAM. The virus isolation positive rate in the five cell types is presented in Table 1.

Next, the differences of viral titers of isolates were analyzed between cell types and virus genotypes (Figure 1). PAM showed the highest average titers compared with the other cell types. Concerning the virus titers obtained with the five different cell types, significant differences were observed between PAM and PK15^S10-CD163^ (PRRSV1: *p* = 0.01 and PRRSV2: *p* = 0.005), PAM and MARC-145^Sn^ (PRRSV1: *p* < 0.0001 and PRRSV2: *p* = 0.005), and PAM and MARC-145 (PRRSV1: p < 0.0001 and PRRSV2: *p* = 0.002). No significant difference was observed between PAM and PK15^Sn-CD163^ (PRRSV1: *p* = 0.74 and PRRSV2: *p* = 0.10). When comparing the mean virus titers obtained with the two different virus genotypes (PRRSV1 vs. PRRSV2), a significant difference was observed with MARC-145 (*p* = 0.022) and MARC-145^Sn^ (*p* = 0.001), whereas this was not observed with PAM (*p* = 0.13), PK15^Sn-CD163^ (*p* = 0.25), and PK15^S10-CD163^ (*p* = 0.17). Further, it was confirmed that PRRSV2 strains compared with PRRSV1 reached a higher titer on MARC-145 cells. PAM, PK15^Sn-CD163^, and PK15^S10-CD163^ cells were more suitable for virus isolation and titration than MARC-145 cells.

### 3.2. Comparison of PRRSV Replication in PAM, PK15^Sn-CD163^, PK15^S10-CD163^, MARC-145^Sn^ and MARC-145

To find correlations between the virus titers obtained with PAM and titers obtained with the other cell types, a linear regression analysis was performed (Figure 2). The *X*-axis represents the virus titers obtained with PAM (“gold standard”) and the *Y*-axis represents the corresponding titers detected with other cell types (PK15^Sn-CD163^, PK15^S10-CD163^, MARC-145^Sn^, and MARC-145). The result of this analysis highlights that the slope for all cells was lower than 1 (0.14–0.77). This is illustrated by the regression line leaning more to the *X*-axis, which indicates that PAM provide more frequently positive results and higher virus titers in contrast to the other cell types.

The R-squared value, a statistical measure denoting how close the data are to the fitted regression line, was determined. Based on the R-squared value, PK15^Sn-CD163^ showed the most similar titration results to that of PAM (PRRSV1: R^2^ = 0.71; PRRSV2: R^2^ = 0.50, *p* < 0.001), followed by PK15^S10-CD163^ (PRRSV1: R^2^ = 0.62; PRRSV2: R^2^ = 0.45, *p* < 0.001), MARC-145^Sn^ (PRRSV1: R^2^ = 0.12, *p* = 0.22; PRRSV2: R^2^ = 0.15, *p* = 0.03), and MARC-145 (PRRSV1: R^2^ = 0.13, *p* = 0.04; PRRSV2: R^2^ = 0.05, *p* = 0.12).

### 3.3. Phylogenetic Analysis and Genome-Wide Association Study between Cell Tropism of PRRSV Isolates and Genomic Sequences

To analyze the potential correlation of genotypes with cell tropism, phylogenetic analyses were performed for classification of the obtained sequences. Based on the full-length sequence phylogenetic analysis (Figure 3), PRRSV1 isolates all belong to genotype 1 subtype 1. One of the isolates (19V93-15) is closely related to the Suvaxyn PRRS MLV vaccine strain. PRRSV2 (Canadian) isolates are mostly distributed in lineage 1, except for isolates 2067408, 2050887, and 2057423, which are closely related to the RespPRRSMLV vaccine strain, belonging to lineage 5 and 2035290, 2018114, and 2018119, which are closely related to the Ingelvac ATP and P129 vaccine strains, both belonging to lineage 8.

We further analyzed the possible residues or motifs in the PRRSV genomes that might be related to the MARC-145 permissiveness. The sequences were divided into two groups based on the titration results: one group containing all the isolates that have shown negative titration results on MARC-145 cells and a second group containing isolates showing positive titration results on MARC-145 and/or MARC-145^Sn^ cells. Genetic associations with either of the aforementioned phenotypes were assessed using the R script (https://github.com/itrus/GWAS-fasta, accessed on 22 February 2015), as previously described [40]. This analysis was carried out separately for PRRSV1 and PRRSV2 isolates to diminish the differences between these genotypes. The results were illustrated with a Manhattan dot plot. Residues with significantly different occurrences in the two subgroups were indicated in red (*p* < 0.001) in the Appendix A. Based on this GWAS, two significant marker sites were identified within the PRRSV2 genome. These two sites are located in GP2 (position 187) and nucleocapsid (position 28). All MARC-145 (MARC-145 and MARC-145^Sn^) permissive isolates (17/17) had a Leucine at position 187 of PRRSV2 GP2 (GP2:187L) as compared with a 187S/T in non-permissive isolates. The majority of MARC-145 permissive isolates (12/17) had an Arginine (R) at position 28 of the N gene, whereas 11 out of 13 isolates of the non-permissive group had a Lysine (K). For PRRSV1, no differences were found. These results were illustrated for the corresponding isolates in the phylogenetic tree in Figure 3.

Sequences of the isolates that showed high virus titers (≥3 log10 TCID_50_/mL) on PAM and sequences of the isolates with low titers (<3 log10 TCID_50_/mL) were also compared with GWAS. Based on the results (Appendix A), one residue was identified for each genotype. For PRRSV1, the residue at position 155 in Nsp11 was found to be significantly different (*p* < 0.01) in the two subgroups: the isolates showing high titers on PAM (7/8) had a Phenylalanine and 9 out of 16 isolates with low titers had a Serine (S). For PRRSV2, one residue in GP5 (at position 78) showed a significant difference between the two subgroups (*p* < 0.001): while 14 out of 16 isolates with a high titer on PAM showed an Isoleucine (I) at position 78, 10 out of 14 isolates with a low titer had a Valine (V). All residues which were identified as potential genetic markers for the cell tropism/titers are shown in Figure 3.

## 4. Discussion

Vaccination is one of the most efficient strategies capable of minimizing economic losses caused by PRRSV. Currently available modified live and inactivated PRRSV vaccines depend on in vitro virus production on MARC-145 cells. In order to grow field isolates of PRRSV to high levels to make autogenous inactivated vaccines, susceptible cell lines are essential. Because of its restricted cell tropism, only a limited number of cell culture options for PRRSV growth are available. In vivo, PRRSV shows a strict preference for certain macrophage subsets, such as porcine alveolar macrophages (PAM) that are considered to be its primary target. These primary porcine cells can be isolated from porcine lungs and directly used for virus isolation and propagation without the requirement of cell adaptation, which is inherent to the in vivo relevance of these cells for PRRSV infection. Apart from PAM, the other cell type known to be permissive to PRRSV is the immortalized monkey kidney cell line MA-104 and its derivatives, such as MARC-145 [22,41]. Currently, the MARC-145 cell line is most commonly used in veterinary diagnostic and research laboratories for PRRSV isolation, propagation, and titration. However, a lot of strains do not grow in MARC-145 cells, especially type 1 strains, therefore limiting their use for virus isolation [42,43]. A recent study demonstrated that only around half of the ZMAC (cell line derived from PAM) obtained isolates are able to grow on MARC-145 cells [44]. Based on our current study, 62.3% of the PCR-positive samples were positive on PAM, while only 27.3% were positive on MARC-145 cells. Thus, we demonstrated that PAM is the most sensitive cell culture for PRRSV isolation. Ideally, PRRSV isolation should therefore be performed in PAM instead of MARC-145. However, due to the primary nature of PAM, they have many disadvantages, including having time-consuming collection, potential risk for contaminations, short lifespan, and batch-to-batch variation. As a result, its widespread use is restricted. To better mimic primary PAM, our laboratory had already successfully established several continuous cell lines expressing dual cell receptors in porcine cell lines, namely a PK15 cell line expressing both Siglec-1 and CD163 (PK15^Sn-CD163^) [28] and a PK15 cell line expressing both Siglec-10 and CD163 [5], as well as MARC-145 cells expressing sialoadhesin (Sn) (MARC-145^Sn^). In the current study, PK15^Sn-CD163^ showed the best performance in virus isolation for both PRRSV1 and PRRSV2 followed by PK15^S10-CD163^. These two cell lines showed the most significant correlation to PAM in field virus isolation. The PRRSV1 isolate 1901_1166 was not growing in PAM and MARC-145 cells but was growing in PK15^Sn-CD163^ and PK15^S10-CD163^, indicating the presence of variants with a tropism for a macrophage subpopulation, different from PAM. Based on virus isolation and titration, PK15^Sn-CD163^ and PK15^S10-CD163^ are good candidates for replacing primary PAM in virus isolation and virus production for the development of autogenous inactivated vaccines.

Based on the isolation rate and virus titers, PRRSV2 strains grew better on MARC-145/MARC-145^Sn^ cells compared with PRRSV1 strains. Some of these strains grew on MARC-145 cells but did not grow on PAM or at a very low titer. These PRRSV2 isolates belong to lineage 5 (isolates 2067408, 2050887, and 2057423 with more than 99% nucleotide homology with the RespPRRSMLV vaccine) and lineage 8 (2035290 with 99.5% identity with the Ingelvac ATP vaccine, and isolates 2018114 and 2018119 with more than 98% identity with the P129 vaccine). Likewise, PRRSV1 isolate 19V93-15 that was not growing on PAM showed 97.0% similarity to the vaccine Suvaxyn PRRS MLV strain. These results indicate that these isolates most probably originated from MARC-145 adapted, attenuated vaccines. This explains why they do not grow (well) in PAM. The frequent isolation of strains that are related to vaccine strains is in line with the wide use of MARC-145 adapted vaccines in the field.

To find the potential residues that are linked to cell tropism and replication efficacy, sequences from 54 field isolates were determined. A genome-wide association study was performed based on the replication efficacy on different cell types. One residue from GP2 (187L) was significantly associated with the MARC-145 growing PRRSV phenotype (*p* < 0.01). In a previous study, we have demonstrated that GP2 amino acid residues at positions 88, 94, and 95 are responsible for PRRSV1 adaptation on MARC-145 cells, pointing towards GP2 as an important structural protein that is involved in the interactions with host cells, more precisely, in the interaction with the CD163 disassembly mediator [40]. A previous study on PRRSV2 demonstrated that the predominant quasispecies in a ZMAC cell (an immortalized macrophage cell line)-prepared virus stock contained a highly conserved N-glycosylation site (NGS) at position 184 in GP2 while this was underrepresented in the MARC-145 cell grown isolates [45]. N184 in GP2 is suggested to enhance virus replication by a stronger interaction of the virus with CD163 (Das et al., 2010; Das et al., 2011). In our current study, the identified residue 187L is just adjacent to the 184N N-glycosylation site. Whether there are interactions between this residue and the 184N glycosylation site requires further investigation. The residue identified in this study (187L) could be an additional essential indicator residue for MARC-145 permissiveness for PRRSV2.

Another important residue in the GP5 protein has been identified between low- and high-level PAM growers. Almost all of the isolates with high titers on PAM had an Isoleucine at residue 78 (78I) of GP5. A previous study of our research team has shown that Siglec-1 is involved in the entry process by interacting with the GP5/M protein complex [46]. PK15^Sn-CD163^ cell permissiveness analysis also showed that residue 78I is significantly correlated to the PK15^Sn-CD163^ permissive phenotype (*p* < 0.001) (data not shown). These results indicate that residue 78 of GP5 could be a potential interaction site with Siglec-1. PRRSV nsp11, a replicative nidoviral uridylate-specific endoribonuclease (NendoU)-like protein, is considered as a major genetic marker that discriminates nidoviruses (*Coronaviridae, Arteriviridae,* and *Roniviridae*) from all other RNA virus families [47]. It plays an important role in viral RNA synthesis and the viral life cycle [48]. A recent study demonstrated that nsp11 can promote the replication of PRRSV, indicating its important in the replicative cycle of PRRSV [49]. PRRSV nsp11 spans residue 3607G -3829E of the polyprotein 1ab (pp1ab) and harbors a conserved 79-amino acid EndoU domain (region between residues 3733L and 3811F). Subdomain A is conserved across the EndoU family and includes three putative catalytic residues: 3735H, 3750H, and 3779K [48]. Our study showed that residue 155 from PRRSV1 nsp11 (corresponding to residue 3760 in pp1a), which is located in the EndoU subdomain A, is potentially linked to the high replication efficiency on PAM (*p* < 0.01). F3760 is present in almost all isolates with a high titer on PAM, whereas it was barely present in the isolates with low titers. Instead, Serine (S3760) is present in most of the isolates from this latter group. 3760F is located between the two catalytic residues 3750H and 3779K. A mutation from phenylalanine (an amino acid with a hydrophobic side chain) to serine (an amino acid with a polar uncharged side chain) could potentially impact the catalytic activity for the residues of 3750H and 3779K and eventually result in a change of virus production. As there are seemingly no black-and-white situations in different phenotypes that might be correlated to the cell tropism, the results obtained from the current study should be confirmed with directed mutagenesis.

With the data presented in our current study, we can make the following conclusions: (i) PAM is the most sensitive cell type for PRRSV1 and PRRSV2 isolation and titration. However, there were also some isolates that only grew on PK15^Sn-CD163^/PK15^S10-CD163^ cells, indicating that some PRRSV variants may have a tropism for a subpopulation of macrophages different from PAM. (ii) There is a clear correlation between the virus titer in PAM and the virus titers in PK15^Sn-CD163^ (*p* < 0.0001) and PK15^S10-CD163^ cells (*p* ≤ 0.0001). (iii) PRRSV2 strains grow much better on MARC-145 cells compared with PRRSV1 strains. Some strains can only grow on MARC-145/MARC-145^Sn^ cells and are related to vaccine strains. (iv) It is feasible to grow PRRSV directly from sera in PK15^CD163-S10^ and PK15^CD163-Sn^ cells, which opens perspectives for future autogenous vaccine development. (v) One site located in the GP2 (187L) of PRRSV2 might be correlated to the cell permissiveness of PRRSV2 in MARC-145 cells, and residue 78I in GP5 of PRRSV2 might be linked to a stronger replication in PAM based on the results of the GWAS. For PRRSV1, no specific residues were identified for MARC-145 cell permissiveness. Residue 155F in nsp11 showed a strong correlation with the replication in PAM.

## Figures and Tables

**Figure 1 vaccines-09-00594-f001:**
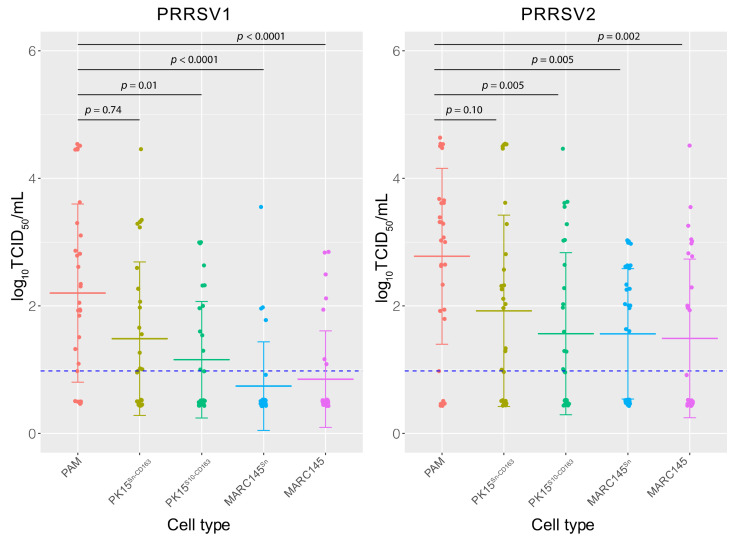
Comparison of virus titers obtained with the five different cell types. Virus titers obtained with each cell line were labeled with different colors. Data are given by dot plot including the mean value of the titers obtained from all the tested samples. The detection limit (0.96 log_10_TCID_50_) was indicated with the blue dashed line. Differences were considered significantly different at *p* < 0.05.

**Figure 2 vaccines-09-00594-f002:**
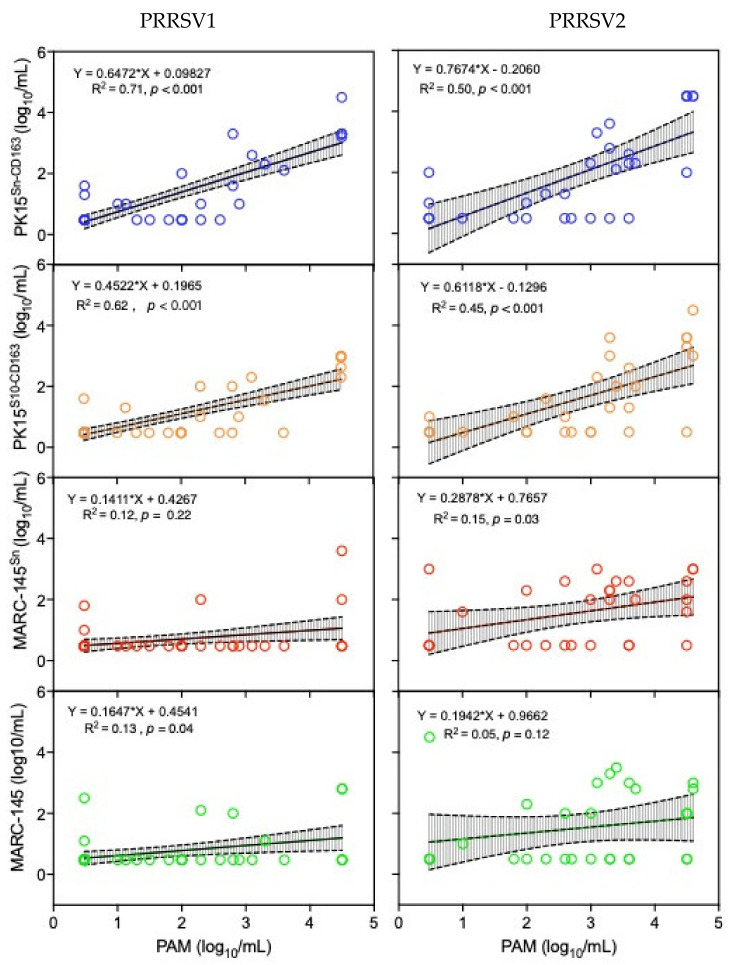
Correlation analysis of type 1 and type 2 PRRSV titration results obtained with PAM and one of the four continuous cell lines (PK15^Sn-CD163,^ PK15^S10-CD163,^ MARC-145^Sn,^ or MARC-145). Each bullet represents the virus titer obtained with PAM (*X*-axis) versus the titer obtained with one of the four continuous cell lines (PK15^Sn-CD163^, PK15^S10-CD163^ MARC-145^Sn^, and MARC-145) (*Y*-axis). The line of linear regression was generated and presented in the figure with a 95% confidence interval (CI_95%_).

**Figure 3 vaccines-09-00594-f003:**
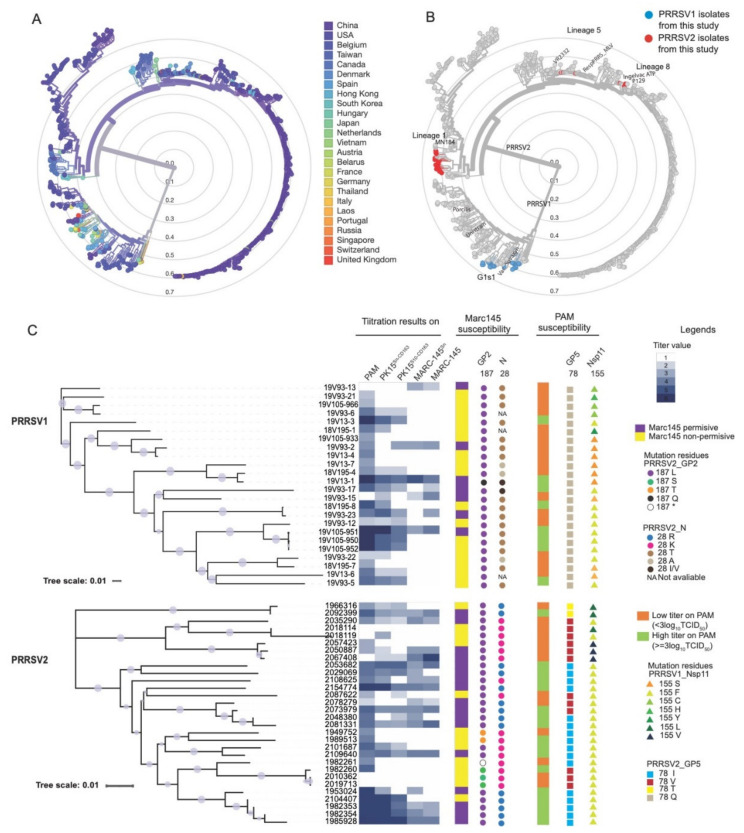
Phylogenetic analysis and the GWAS analysis between genotype and virus-cell tropism for PRRSV1 and PRRSV2 field isolates. (**A**) The phylogenetic tree containing both PRRSV1 and PRRSV2 full-genome sequences with available reference sequences from GenBank, labeled with countries. (**B**) The distribution of the isolates (PRRSV1 and PRRSV2) from the current study in the full-genome phylogenetic tree. (**C**) Phylogenetic and phenotypic correlation analysis. The phylogenetic tree was constructed with an IQ-Tree. The tree and the titration data set were combined. The bootstrap values above 70 were indicated with blue circles in each node of the tree. The heatmap in the first column, besides the tree, represents the titration data tested in the five cell lines, as indicated. The second column represents the residues that are potentially linked to MARC-145 permissiveness. The third column shows the residues that are potentially linked to preferences for PAM. Legends for all the symbols are listed in the figure. The asterisk (*) represents a stop codon in the corresponding residue.

**Table 1 vaccines-09-00594-t001:** Summary of virus isolation rate in PAM, PK15^Sn-CD163^, PK15^S10-CD163^, MARC-145, and MARC-145^Sn^.

Virus	No. qPCR^+^ Sera	No. of Samples Positive in … Cells (Positive Rate %)				
		PAM	PK15^Sn-CD163^	PK15^S10-CD163^	MARC-145^Sn^	MARC-145
PRRSV1	47	23 (48.9)	17 (36.2)	13 (27.7)	5 (10.6)	7 (14.9)
PRRSV2	30	25 (83.3)	19 (63.3)	17 (56.7)	17 (56.7)	14 (46.7)
All	77	48 (62.3)	36 (46.8)	30 (39.0)	22 (28.6)	21 (27.3)

## Data Availability

Data is contained within the article. Reported results can be found in Appendix A.

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
