# Peer review of "Comparison of Primary Virus Isolation in Pulmonary Alveolar Macrophages and Four Different Continuous Cell Lines for Type 1 and Type 2 Porcine Reproductive and Respiratory Syndrome Virus"

_vaccines, 2021, doi:10.3390/vaccines9060594_

Round 1

Reviewer 1 Report

PRRSV is and endemic pig pathogen that affects the swine industry worldwide. In this paper, Xie et al have characterized the growth of PRRSV1 and PRRSV2 isolates in fourth cell lines recombinantly expressing different viral receptors in comparison with the growth in the natural host porcine alveolar macrophages. They conclude that PRRSV1 and PRRSV2 may optimally grow in PK15-derived cells.  Moreover, they have identified residues in three viral proteins that might be involved in viral tropism.

The work is potentially of interest for PRRSV isolation and vaccine production. Minor changes should be addressed before publication. 

Abstract:

  • Lines 36 and 37: it should read GP2:187L and GP5:78I respectively instead of GP2:178L and GP5:87I

Introduction:

  • The authors conclude that some residues from GP2, N and GP5 viral proteins are involved in viral tropism. They should include in the introduction section a brief description of the role of these proteins in the life cycle of the virus.

Results:

  • Through the text, two different names are used for the recombinant cell line derived from PK15 that expresses Siglec-1 and CD163: PK15Sn-CD163 and PK15S1-CD163; the same name should be used.
  • Table 1: The title should read “Summary of virus titration” instead of isolation.
  • Lines 185 and 186: According to data presented in Fig. 1, the p value are 0.74 for PRRSV1 and 0.10 for PRRSV2 respectively.
  • Supplementary Figure 1: The figure must be improved, as data are difficult to see; moreover, a legend should be included. Residues 188 and 318 are not clearly indicated in Sup. Fig.1B.
  • The authors stated that all MARC-145 permissive isolates had a leucine at position 187 of GP2 protein (Line 300). Nevertheless, according to data presented in Supplementary Figure 1, some of the permissive isolates had an Isoleucine residue. This must be corrected in the text as well as in legend to Figure 3 (right panel).
  • For the analysis of N gene, it is stated that the majority of MARC-145 permissive isolates had an arginine (12/17, 71 %) (lines 301-302). In data presented in supplementary Figure1, it seems that 9 isolates have an arginine (53 %), 5 a lysine and 3 and Asn.
  • Legend to Figure 3 (right panel), line 339 “High titer on…” PAM is missed
  • It would be interesting to carry out virus-cell binding assays to analyze the role of the residues GP2:187, N:28 and GP5:78 in viral entry.

Author Response

please find in the attachment.

Reviewer 2 Report

PRRSV 1 and PRRSV 2 field strains poses significant challenge for primary isolation in cell culture because one cell type is not fully susceptible for virus isolation from PCR + serum samples. This is an important cell line evaluation work for PRRSV field strains susceptibility to help virus isolation and vaccine production.

  1. The researchers could include all 5 cell lines in Heat Map fig 3 C to demonstrate susceptibility to the PRRSV strains used, or additional heat map table can be made to virus indicate isolate specific susceptibility to all 5 cell lines. This would help to determine if all 5 cell lines should be tried for virus isolation. Example, is there any field virus isolate that was not isolated in PAM or PK15 and was isolated in MA or derivative cell lines? This comment is also in context to table 1 in which the total number of isolations is shown and the individual isolate distribution on success or growth in different cell types is missing. The data needs elaboration or dissection.
  2. Does authors have a final recommendations to other laboratories that one particular cell type such as engineered PK15 S1-CD163 or PK15 S10-CD163 or simultaneous engineered PK15 and MA derivative cell types panel can be used to attempt for virus isolation in absence of PAM? Are there virus isolates that can grow on MA derivative cell but not on PK15 derivatives?

Author Response

Reviewer 2:

  • The researchers could include all 5 cell lines in Heat Map fig 3 C to demonstrate susceptibility to the PRRSV strains used, or additional heat map table can be made to virus indicate isolate specific susceptibility to all 5 cell lines. This would help to determine if all 5 cell lines should be tried for virus isolation. Example, is there any field virus isolate that was not isolated in PAM or PK15 and was isolated in MA or derivative cell lines? This comment is also in context to table 1 in which the total number of isolations is shown and the individual isolate distribution on success or growth in different cell types is missing. The data needs elaboration or dissection.

Response:  We would like to thank the reviewer for his/her suggestions. Actually, we have included the heatmap with the virus titers obtained with the five cell types to show the susceptibility of these five cell types in Figure 3C.  Indeed, there were only a few virus strains that were not isolated in PAM but were isolated in MARC-145 cells. These have been shown in Table S1: Titration results for PRRSV1 on PAM, PK15Sn-CD163, PK15S10-CD163, MARC-145Sn and MARC-145. This has also been discussed in the revised manuscript from Line 377-387: "Some of these strains grew on MARC-145 cells but did not grow on PAM or at a very low titer. These PRRSV2 isolates belong to lineage 5 (isolates 2067408, 2050887 and 2057423 with more than 99% nucleotide homology with the RespPRRSMLV vaccine) and lineage 8 (2035290 with 99.5% identity with the Ingelvac ATP vaccine and isolates 2018114 and 2018119 with more than 98% identity with the P129 vaccine). Likewise, PRRSV1 isolate 19V93-15 that was not growing on PAM showed 97.0% similarity to the vaccine Suvaxyn PRRS MLV strain. These results indicate that these isolates most probably originated from MARC-145 adapted, attenuated vaccines. This explains why they do not grow (well) in PAM. The frequent isolation of strains that are related to vaccine strains is in line with the wide use of MARC-145 adapted vaccines in the field.” 

  • Do the authors have a final recommendation to other laboratories that one particular cell type such as engineered PK15 S1-CD163 or PK15 S10-CD163 or simultaneous engineered PK15 and MA derivative cell types panel can be used to attempt for virus isolation in absence of PAM? Are there virus isolates that can grow on MA derivative cell but not on PK15 derivatives?

Response: Based on our results, PAM is the most sensitive cell type for PRRSV1 and PRRSV2 isolation and titration. However, there were also some isolates that only grew on PK15Sn-CD163/PK15S10-CD163 cells, indicating that some PRRSV variants may have a tropism for a subpopulation of macrophages different from PAM. PK15Sn-CD163 cells that are most closely related to PAM have the best performance in virus isolation and titration compared to the other continuous cell lines. The virus strains that are only growing on MARC-145 are most likely to vaccine strains. This has been discussed in the Discussion part.

Reviewer 3 Report

Comments to the Authors of manuscript number: vaccines-1220358 entitled “Comparison of primary virus isolation in pulmonary alveolar macrophages and four different continuous cell lines for type 1 and type 2 porcine reproductive and respiratory syndrome virus”.

1.L 19- abbreviation should be explained

  1. L 57 - abbreviation should be explained
  2. L 61- reference should be added
  3. L76- 78 this part should be rephrases. Authors should omit the term of “our”. Moreover, there should be given a reference
  4. L98- it should be explained if serum positive was from Belgian swine
  5. L 102 – the number of samples should be given
  6. L 107 – the company of the monoclonal antibody should be given
  7. L 114 - the location of the laboratory should be given
  8. The company of all biochemicals should be described shortly. It is good described they in one short paragraph.
  9. In table 1, there is given the number of sera. It is the whole number of sera used or there were more samples, and what criteria for sample was used?
  10. Why there is a need to look for other cells lines? It should be explained.

Author Response

Reviewer 3:

Comments to the Authors of manuscript number: vaccines-1220358 entitled “Comparison of primary virus isolation in pulmonary alveolar macrophages and four different continuous cell lines for type 1 and type 2 porcine reproductive and respiratory syndrome virus”.

1. L 19 - abbreviation should be explained

Response:  PRRSV (Porcine Reproductive and Respiratory Syndrome Virus) has been explained in the revised manuscript.

2. L 57 - abbreviation should be explained

Response:In the revised manuscript, ORF (Open Reading Frame) has been written in full.

3. L 61 - reference should be added

Response: Reference has been added in the revised manuscript.

4. L76 - 78 this part should be rephrases. Authors should omit the term of “our”. Moreover, there should be given a reference

Response:This sentence has been rephrased and the reference has been included.

Previously, we have established two stably transfected cell lines: one expressing both Siglec-1 and CD163 (PK15Sn-CD163) and one cell line stably expressing both Siglec-10 and CD163 (PK15S10-CD163) [5, 28]

5. L98 - it should be explained if serum positive was from Belgian swine

Response: It has been rephased, PRRSV1 serum positive samples were collected…

6. L 102 - the number of samples should be given

Response: The number has been added in the revised manuscript

In total 77 serum samples …

7. L 107 - the company of the monoclonal antibody should be given

Response: The antibody was produced in our laboratory. We have cited the reference that described the antibody.

8. L 114 - the location of the laboratory should be given

Response: The location has been added in the revised manuscript. The location is Merelbeke, Belgium.

9. The company of all biochemicals should be described shortly. It is good described they in one short paragraph.

Respondse: we have added all the companies besides the biochemicals used in our manuscript.

10. In table 1, there is given the number of sera. It is the whole number of sera used or there were more samples, and what criteria for sample was used?

Response: The main purpose of our manuscript is to compare the primary isolation of PRRSV in different cells and to find the potential link between genotype and phenotype. For the sample collection, we only took the PRRSV (both PRRSV1 and PRRSV2) PCR positive samples. Thus, for successful isolation of virus, our only criterium was that the samples should be PRRSV PCR positive.

11. Why there is a need to look for other cells lines? It should be explained.

Response: We have explained this in our introduction (Line 104-111) and also in the discussion (Line 340-355).

Round 2

Reviewer 3 Report

I have no comments.